# Vibrational Spectral Analysis of Natisite (Na_2_TiSiO_5_) and its Structure Evolution in Water and Sulfuric Acid Solutions

**DOI:** 10.3390/ma14092259

**Published:** 2021-04-27

**Authors:** Fancheng Meng, Yahui Liu, Lina Wang, Desheng Chen, Hongxin Zhao, Yulan Zhen, Jing Chen, Tao Qi

**Affiliations:** 1National Engineering Laboratory for Hydrometallurgical Cleaner Production Technology, Institute of Process Engineering, Chinese Academy of Sciences, Beijing 100190, China; fcmeng@ipe.ac.cn (F.M.); yhliu@ipe.ac.cn (Y.L.); dshchen@ipe.ac.cn (D.C.); hxzhao@ipe.ac.cn (H.Z.); ylzhen@ipe.ac.cn (Y.Z.); tqi@ipe.ac.cn (T.Q.); 2State Key Laboratory of Complex Nonferrous Metal Resources Clean Utilization, Kunming University of Science and Technology, Kunming 650093, China; 3Institute of Nuclear and New Energy Technology, Tsinghua University, Beijing 100084, China; jingxia@tsinghua.edu.cn

**Keywords:** natisite, vibration spectroscopy, structure evolution, sodium ion, aqueous solution

## Abstract

Natisite (Na_2_TiSiO_5_) is a layered sodium titanosilicate containing TiO_5_ square pyramids. The structure evolution of natisite in water and acid solutions is the basis for its potential applications. With Na_2_SiO_3_ as the silicon source, natisite with the shape of the square sheet was selectively prepared from the hydrothermal method with 14.3 mol/L NaOH solution at 240 °C. Natisite has 20 Raman active modes and 22 infrared active modes from the first-principles calculations within density functional theory, and the calculated Raman and infrared spectra agree well with the experimental ones. The characteristic Raman peak at 844 cm^−1^ is caused by the symmetric stretching of the apical Ti–O bond in the TiO_5_ unit, assigning to *A*_1g_ and *B*_2g_ modes. Natisite remains relatively stable in water with a sodium leaching percentage of lower than 6%. When washing with sulfuric acid solutions, the interlayer spacing of natisite is reduced due to the extensive removal of sodium ions, and an intermediate composed of SiO_4_ and newly formed TiO_6_ units may be formed. Moreover, after washing with water and acid solutions, 95.5%, 63.4%, and 35.2% of Na, Si, and Ti in natisite can be leached in total, respectively, resulting in the structural disintegration of natisite.

## 1. Introduction

In the last decades, numerous microporous or layered titanosilicates containing the mixed polyhedral framework built by tetrahedral Si fragments and Ti units with six-fold or rarely five-fold coordination have been synthesized due to their interesting properties [1,2,3]. As a member of the titanosilicate materials, Na_2_TiSiO_5_ is attracting increasing attention, and it has three polymorphs [4]. The first one is the high-temperature structure that can only exist at above 772 °C [5] and is considered as a tetragonal structure with the space group of *P*42/*m*. The second one is paranatisite, which has an orthorhombic structure (*Pmc*21) with *a* = 9.181(2) Å, *b* = 4.800(1) Å, *c* = 9.811(2) Å, *Z* = 2 [6]. Due to the vacancy of Ti and the substitution of O by OH, paranatisite was originally refined in the space group *Pmma* with the formula of Na_8_Ti_3_._5_O_2_(OH)_2_(SiO_4_)_4_, which was thought to be a low-temperature orthorhombic modification of natisite [7]. Paranatisite has two Ti ion sites, namely isolated TiO_5_ square pyramids and TiO_6_ octahedron, making it a good candidate for phosphor material with a white emission band from 400 nm to 650 nm, and doping Cr^3+^ could effectively regulate its luminescence properties [8,9]. The third polymorph is also the tetragonal structure (*P*4/*nmm*), named natisite, with the lattice parameters of *a* = *b* = 6.480 Å, *c* = 5.090 Å, *Z* = 2 (JCPDS 01-086-1615). Natisite is unique with its rare five-coordinated titanium, and its crystal structure contains layers of TiO_5_ square pyramid and SiO_4_ tetrahedron joined at corners and separated by layers of sodium ions [10]. The phase transformation between these three polymorphs has been reported previously [5,11,12], among which natisite is a more stable phase in NaOH solutions at ≥200 °C. Because of its layered structure with sodium ions, natisite is expected to have ion-exchange property, which has aroused our research interest.

Natisite could be used as sorbent, and the inclusion of zirconium has increased its exchange capability and made it a promising ion-exchanger for the treatment of nuclear wastewater [13]. Natisite shows an acceptable hydrogen adsorption capacity, and the incorporation of vanadium can improve its storage capacity [14]. Natisite can be used as the anode of lithium-ion batteries with high capacity, low working potential, and excellent rate performance [15]. Natisite also exhibits the ability to store sodium ions, but the diffusion kinetics of sodium ions limits its application as a sodium-ion battery anode [16]. Natisite could be used as a transition intermediate in synthesizing the layered titanosilicate AM-4 [17], and it is also the thermal treatment product of sitinakite [13]. In the extractive metallurgy of titanium, natisite with the morphology of the square sheet was formed as the byproduct of the alkaline roasting and hydrothermal conversion of titanium-containing raw minerals [18,19]. Similar to the behavior of sodalite in the process of alumina production by the Bayer process [20], sodium ions in natisite are difficult to leach out in the water, which could lead to severe Na_2_O loss in the process [21]. As a layered sodium titanosilicate, it is essential to clarify how much and how sodium ions in natisite could be exchanged or leached in water and acid solutions, which is also helpful to understand the ion-exchange behavior of natisite as an adsorbent.

Natisite can be prepared by both solid-state reaction and hydrothermal methods [11]. By adjusting the molar ratios of Na_2_O/TiO_2_ and TiO_2_/SiO_2_ in the hydrothermal process, Stanislav Ferdov selectively synthesized natisite with truncated bipyramidal, disk-shaped, and pillowlike microstructures [22,23], in which SiO_2_ was used as the silicon source [23,24]. A double-walled natisite nanotube was prepared by an alkaline hydrothermal process with cetyltrimethylammonium bromide as an assistant [2]. Previous studies have shown that the hydrothermal method is a reasonable approach to prepare natisite with controllable morphology.

The structure of natisite has been characterized by many spectroscopic methods, such as FT-IR [11,25], Raman [24,26], and NMR [1,24], which provide much detailed structural information. However, previous studies only speculated the vibration of Ti‒O and Si‒O bonds of the Raman peaks at around 850 and 900 cm^−1^ [24,26], which are not complete and need further experimental or theoretical verification. The vibrational modes of Raman peaks and infrared absorption bands of natisite still remain unassigned. Recently, interest in using the first-principles calculations within density functional theory to explore the vibrational modes of materials has deepened [27,28,29]. With the assistance of the calculated atom displacements, the correct assignment of vibrational modes provides a strong basis to study the structural evolution of natisite.

In this study, natisite with the shape of a square sheet was selectively prepared from TiO_2_ in NaOH solution using the hydrothermal method with Na_2_SiO_3_ used as the ionic-type silicon source, which could promote the completion of the hydrothermal reaction. The structure of natisite was geometrically optimized, and the first-principles calculations were carried out to explore the assignment of its vibration modes for the first time. Its structure evolution in removing sodium ions by washing it with water and sulfuric acid solutions was investigated systematically.

## 2. Experiments and Calculations

### 2.1. Preparation of Natisite

NaOH (purity ≥ 98.0%), TiO_2_ (purity ≥ 99.0%, anatase phase, particle size 0.2~0.3 μm), Na_2_SiO_3_·9H_2_O (purity ≥ 98.0%) and deionized water (Milli-Q, Millipore, Billerica, MA, USA) were used to prepare natisite by hydrothermal method. All these reagents were analytically pure and purchased from Sinopharm Chemical Reagent Co., Ltd (Shanghai, China). NaOH, and Na_2_SiO_3_·9H_2_O were first dissolved in deionized water, and then TiO_2_ powder was added and dispersed evenly. The molar ratio of NaOH, TiO_2_ and Na_2_SiO_3_ was 7.0:1.0:1.1 in preparation. Different amount of deionized water was used to adjust the concentration of NaOH. Then, the mixture was poured into a nickel autoclave and sealed, which is mechanically stirred to keep it in suspension. Then, the autoclave was heated to the selected temperature at a heating rate of 5 °C/min and thermostatically controlled within ±1 °C. Once the temperature was reached, the reaction was maintained at autogenous pressure of the system for six hours. After the reaction was finished, the mixture in the autoclave was cooled and filtered. The solid product was sampled, washed with analytically pure ethanol, and dried at 80 °C in an electric blast drying oven overnight for further analysis.

### 2.2. Washing Natisite with Water and Sulfuric Acid Solutions

Natisite with the shape of a square sheet as prepared in Section 2.1 was first washed with deionized water with the liquid/solid mass ratio of 3:1 at 60 °C for 40 min. This process was repeated four times. After this, it was washed with sulfuric acid solutions with the liquid/solid mass ratio of 3:1 at 25 °C for 40 min, and this process repeated four times with the pH values of the washing solutions of 6.5, 4.5, 1.0, and −0.4, respectively. Sulfuric acid (purity ≥ 98.0%, Beijing Chemical Works, Beijing, China ) and deionized water were used to ajust the pH values. To further investigate the behavior of natisite in a more acidic solution, natisite was directly leached with 40 wt % H_2_SO_4_ solution with the H_2_SO_4_/Na_2_TiSiO_5_ molar ratio of 3:1 at 65 °C for four hours, followed by washing the solid residue with water at 25 °C for 40 min. After each test, the mixture was filtered, and the composition of filtrates was analyzed to calculate the leaching percentages of Na, Si and Ti. The solids were sampled, dried at 80 °C overnight for further structural studies. These serial treatment procedures for natisite and the names of the obtained solid samples are listed in Table 1.

### 2.3. Characterization

The solid phases were identified using X-ray diffraction (XRD, Empyrean, PANalytical, Almelo, the Netherland) with Cu K*α* radiation, 40 kV, 40 mA, and the scanning range from 5° to 90°. The chemical compositions of the solid samples and solutions were determined via inductively coupled plasma optical emission spectrometry (ICP-OES, iCAP6300, Thermo Fisher, Waltham, MA, USA). Raman spectra were recorded using a Raman spectrometer (LabRAM HR800, HORIBA Jobin Yvon, Paris, France) with the pulsed exciting light (514.5 nm). Fourier-transform infrared (FT-IR, Spectrum GX, PerkinElmer, Waltham, MA, USA) spectra were acquired using KBr tablets. The micromorphology and chemical analysis were realized by the scanning electron microscopy (SEM, JSM-6700, JEOL, Akishima, Tokyo, Japan) combined with energy-dispersive X-ray spectroscopy (EDS, INCAX-MAX, Oxford Instruments, Abingdon, UK).

### 2.4. Computational Details

The CASTEP package was used to carry out the plane-wave pseudopotential calculations using density functional theory [30]. The crystallographic information file of natisite Na_2_TiSiO_5_ from H. Nyman [10] was used in geometry optimization. The exchange-correlation potential of Ceperley and Alder parameterized by Perdew and Zunger (CA-PZ) [31,32] was used in the local density approximation (LDA) as the electron–electron interactions in all calculations, and norm-conserving pseudopotentials were adopted [33]. The electronic valence configurations for Na, Ti, Si, and O were Na-2*p*^6^3*s*^1^, Ti-3*d*^2^3*p*^6^4*s*^2^, Si-3*s*^2^3*p*^2^, and O-2*s*^2^2*p*^4^, respectively. All integrals in reciprocal space were evaluated with the Monkhorst–Pack [34] 1 × 1 × 1 sampling. Optimization of the unit cell was conducted under the Broyden–Fletcher–Goldfarb–Shanno (BFGS) minimization scheme [35]. A starting Hessian was recursively updated in the BFGS scheme. A plane-wave basis set cutoff energy of 830 eV and a total energy convergence tolerance of 5 × 10^−6^ eV/atom were used in all calculations. After geometry optimization, the IR absorption intensity was obtained by calculating the phonons at the *Γ* point (***k*** = 0), and the Raman spectrum of natisite was calculated based on the Raman effect of inelastic scattering of monochromatic light [36].

## 3. Results and Discussion

### 3.1. Selective Preparation of Natisite

Although the synthesis of natisite has been studied [22,23,24], the effects of temperature and NaOH concentration remain unclear. Figure 1 shows the XRD patterns of the hydrothermal products obtained from different temperatures and NaOH concentrations. Figure 1a indicates that the raw material TiO_2_ disappears gradually with the increase of temperature in 10 mol/L NaOH solution, and only the paranatisite phase was observed at 180 °C, while both natisite and paranatisite were formed at 210 °C. The strongest reflection peaks of natisite and paranatisite are 32.70° and 32.57°, respectively, which is too close to be distinguished. Figure 1b indicates that paranatisite was observed in 6 mol/L and 10 mol/L NaOH solutions, while pure natisite phase was obtained in 14.3 mol/L NaOH solution at 240 °C. It is proved high alkali favors the formation of natisite, consistent with the previous work [37]. It may be because paranatisite is unstable, and it is converted into natisite under certain conditions [12], such as higher temperatures and NaOH concentrations.

As shown in Figure 2, the morphologies of hydrothermal products became more regular with the increase of reaction temperature in 10 mol/L NaOH solution, and a few fibrous materials were formed at 210 °C, while the lamellar aggregates were obtained at 240 °C. However, according to Figure 1, the natisite product was not pure due to the low conversion of TiO_2_ at this NaOH concentration. At 240 °C, the product was irregular agglomerate in 6 mol/L NaOH solution, and when the NaOH concentration was increased to 14.3 mol/L, pure natisite phase with the shape polish square sheet was obtained. Its average molar ratio of Na:Si:Ti is in fair agreement with the composition of Na_2_TiSiO_5_ in EDS analysis. Therefore, natisite in Figure 1f is optimal and used in the following study.

### 3.2. Structure and Vibrational Spectral Analysis of Natisite

The atomic sites of natisite Na_2_SiTiO_5_ from geometry optimization are shown in Table 2, and the optimized lattice parameters are *a* = *b* = 6.3853 Å, *c* = 5.0137 Å. The ball-and-stick and polyhedral representations of natisite and O = Ti(O–Si)_4_ unit are shown in Figure 3.

There is only one site for Ti, Si and Na and two sites for O, which indicates that this structure is highly symmetrical. All four corners of the SiO_4_ tetrahedron are shared with four lateral oxygen of the titanium polyhedron. The fifth oxygen of the TiO_5_ pyramid is apical oxygen, which points to the interlayer space. The bond length between the apical oxygen (O^a^) and titanium is 1.69 Å, much shorter than 1.94–1.98 Å of Ti–O bonds in anatase and rutile, which indicates that Ti–O^a^ bond in natisite is very strong and could be treated as Ti = O [26]. As shown in Figure 3, the distances between O^a^ and the surrounding Ti and Si are too large to form chemical bonds. Furthermore, unlike the rarely observed TiO_5_ trigonal bipyramids in *γ-*Na_2_TiO_3_, and TiO_5_ asymmetric square pyramids are formed here, a common form for Ti(IV) of five-fold coordination [23,38]. Six-coordinated sodium ions are located in the layer built by TiO_5_ and SiO_4_ units, and they may be extracted from natisite via migrating through the layer space or breaking down the layered structure.

Table 3 presents the predicted vibration frequencies of normal modes with the respective irreducible representations and the assignment of IR and Raman active modes of natisite obtained from the first-principles calculations. The point group of natisite is *D*_4*h*_, and there are 20 Raman active modes and 22 IR active modes according to the group theory analysis of eigenvectors with *Γ_Raman_* = 12*E_g_* + 4*A*_1*g*_ + 3*B*_2*g*_ + *B*_1*g*_ and *Γ_IR_* = 16*E_u_* + 6*A*_2*u*_. However, there are only 14 Raman frequencies and 14 IR frequencies due to the existence of six *E_g_* modes and eight *E_u_* modes corresponding to two symmetric vibrations at the same frequency. The raw vibrational output file describing the atomic displacements is provided in the Appendix A.

As shown in Figure 4a, the calculated Raman spectrum is in good agreement with the experimental one. The Raman peak at 889 cm^−^^1^ is assigned to *E*_g_ mode corresponding to the symmetric stretching of Si–O bonds. The intense and sharp peak at 844 cm^−1^ (exp.) is due to the symmetric stretching of the Ti–O^a^ bond in the TiO_5_ square pyramids assigning to the combination of *A*_1g_ and *B*_2g_. The assignments of the above two Raman peaks are consistent with the previous studies [24,26]. As a contrast, the symmetric stretching of the apical Ti–O bond of the TiO_5_ trigonal bipyramids in *γ*-Na_2_TiO_3_ lies at 803 cm^−^^1^ [39]. Moreover, the apical Ti–O bond in natisite is much shorter than that in TiO_4_ and TiO_6_ units, leading to the Raman vibration of Ti–O bonds locating at high frequencies of 800~900 cm^−1^ [26]. The peaks in the range of 400~800 cm^−1^ all involve the vibrations of Ti–O^b^ bonds. The vibration at 522 cm^−1^ is mainly the antisymmetric vibration of the Ti–O–Si bond in the *x–o–z* and *y–o–z* planes, basically perpendicular to the *z*-axis direction, which causes the change of layer spacing. The vibration peak at 399 cm^−1^ is the asymmetric stretching vibration of the Ti–O^b^ bond, and the vibration direction is parallel to the *x–o–z* plane, which also causes the change of layer spacing. The peak at 268 cm^−1^ is contributed by the swing of the Ti–O^a^ bond, assigning to *E_g_* mode. The movement of the SiO_4_ unit along the *z*-axis leads to the vibration at 185 cm^−1^ (exp.), assigning to *B*_2*g*_ mode. Calculation results indicate that the vibration of the Na–O bond produced very weak peaks below 161 cm^−1^, which are not observed in the experimental spectrum.

Figure 4b shows that the calculated infrared spectrum is also well matched with the experimental spectrum of natisite. The absorption bands at 986, 899, and 854 cm^−1^ are attributed to the symmetric stretching vibration of Si–O bond (*A*_2*u*_ mode), the antisymmetric stretching vibration of Si–O bond (*E_u_* mode), and the stretching of the apical Ti–O bond in the TiO_5_ square pyramids, respectively. There are two characteristic sharp IR absorption bands at 726 and 625 cm^−1^, representing the internal modes of TiO_5_ and SiO_4_ structural units, respectively [11], consistent with our calculation result (assigning to the *A*_2*u*_ and *E_u_* modes). Moreover, the antisymmetric stretching vibration of Ti–O^b^ bonds (*E_u_* modes) may lead to the IR absorption at 421 cm^−1^.

### 3.3. Structure Evolution of Natisite in Water and Sulfuric Acid Solutions

#### 3.3.1. ICP Results and XRD Analysis

In this study, natisite was washed with water four times, then washed with the sulfuric acid solution four times, as described in Section 2.2. The leaching percentages of Na, Si and Ti in the washing process are presented in Figure 5. It shows that the leaching percentage of sodium ions remains lower than 6% even after the fourth water washing, while 95.7% of sodium ions can be leached out after the fourth acid washing. The leaching trend of Si is similar to that of Na at pH > 6.5 and becomes much smaller than Na at lower pH values. Ti cannot be leached out until the pH value of aqueous solutions is lower than 1.0, and the leaching percentage of Ti is only 35.2% when almost all Na are leached out after the fourth acid washing.

XRD patterns of the solid samples obtained from washing natisite with water and sulfuric acid solutions for different times are shown in Figure 6. The enhancement of the peak at 17.36° (crystal plane 001) was observed and reached the maximum intensity at the first acid washing (pH = 6.5), which may be attributed to the enhancement of the periodicity in the crystal plane (001) caused by the leaching of sodium ions from the layer.

The peaks of crystal planes (001) and (201) of natisite gradually disappeared in the process of acid washing, which indicates that the layer structure may break down when a large proportion of sodium ions are leached out in sulfuric acid solutions. It is interesting that the XRD peak at 27.46° of crystal plane (200) is intensified gradually in the washing process, which is very close to the characteristic peak of rutile at the crystal plane (110), and this sharp peak is quite different from the broader peak of the rutile phase hydrolyzed from the titanium-containing solution [40]. To reveal the origin of this intensified peak, the solid residue of natisite from the fourth acid washing was calcined at 700 °C, and its XRD pattern Appendix A and Raman spectrum in Appendix A both show that its calcined product belongs to anatase. It suggests that an intermediate containing titanium was formed after the removal of sodium ions, which exhibits a diffraction peak at 27.46°.

Before the disintegration of the layered structure of natisite in sulfuric acid solutions, Na^+^ that was originally located between the layers exchanges with H^+^ in the solution, which decreases the interlayer distance due to the smaller diameter of H^+^. With the removal of sodium ions and the reduction of interlayer distance, the apical O atom in the one TiO_5_ pyramid may have the possibility to connect with the central Ti atom in another adjacent TiO_5_ pyramid to form a more stable TiO_6_ octahedron. After rebuilding and refining with the XRD data, an intermediate structure with the space group of *Cmma* and the lattice parameter of *a* = *b* = 9.1983 Å, *c*= 4.0983 Å, and *Z* = 4 and was proposed, and its crystallographic information file is provided in the Appendix A. This intermediate H_2_TiSiO_5_ is constructed by TiO_6_ and SiO_4_ units with H^+^ located in the tunnel space, and the average Ti–O bond length is 2.025 Å. Its simulated XRD pattern in Figure 7 agrees well with the experimental data, which indicates that a similar intermediate structure could be formed in the structure evolution process.

#### 3.3.2. FT-IR and Raman Spectra Analysis

The FT-IR spectra of the solid samples from washing natisite with water and sulfuric acid solutions are shown in Figure 8. These two characteristic sharp IR absorption bands of natisite at 725 and 625 cm^−1^ have no apparent changes in water washing, gradually weaken in the first three acid washings, and disappeared after the fourth acid washing. It indicates that the layer built by TiO_5_ and SiO_4_ units is stable in water and finally disintegrates in acid washing. The bands assigned to the stretching vibrations of O–H groups (at about 3430 cm^−1^) and the bending vibrations of H–O–H in H_2_O molecules (at about 1630 cm^−1^) are also observed [41,42]. With the disintegration of the structure, many titanyl clusters and silicate materials are supposed to be formed, which is proved by the stretching of apical Ti–O bonds of TiO_2_ clusters at 954 and 1054 cm^−1^ [43] and the Si–O stretching of Q^0^ silicate near 1200 cm^−1^ [44] shown in Figure 8.

In another treatment process, natisite was directly leached with 40 wt % H_2_SO_4_ solution, and almost all sodium and titanium in natisite could be leached. The FT-IR spectrum of its solid residue (sample AX) in Figure 8 exhibits the characteristic absorption peaks of silica gel at 1100, 970, 801 and 470 cm^−1^ [45], indicating a total disintegration of natisite structure and the formation of silica gel in this process.

Raman spectra of the solid samples from washing natisite with water and sulfuric acid solutions are shown in Figure 9. It shows that in the process of water washing and the first two acid washing, the intensities of Raman peaks of natisite are weakened gradually, especially for the vibration peaks of Si–O bonds at 523 and 889 cm^−1^. Combined with ICP analysis in Figure 5, it can be preferred that the partial leaching of Si leads to the weakening or broadening of the Si–O vibration peaks. Moreover, the vibration peak at 269 cm^−1^ caused by in-plane swing of Ti–O^a^ bond shifts to 285 cm^−1^, which proves the decrease of the interlayer distance during acid washing. The Raman peak at 844 cm^−1^ caused by the symmetrical stretching vibration of Ti–O^a^ bond pointing to the interlayer continuously weakened during acid washing and finally disappeared, while a new vibration peak at 813 cm^−1^ appeared, indicating the change of the TiO_5_ pyramid. Furthermore, the Raman vibration peak at 813 cm^−1^ just falls within the vibration range of Ti–O bond inTiO_6_ octahedron, which is consistent with the speculation of the formation of an intermediate structure containing TiO_6_ units in Section 3.3.1.

#### 3.3.3. SEM Analysis

The morphological changes of natisite in water and sulfuric acid solutions were compared, as shown in Figure 10. During water washing, the originally smooth surface of natisite particles gradually became rough with the appearance of a fluffy substance layer. During acid washing, the particle surface gradually evolved into a floccule, and the regular shape and sharp edges of natisite particles disappeared. Finally, the particle geometry of natisite was totally destroyed. Moreover, obvious lamellar structures can be observed in samples A3 and A4. The above macroscopic changes in morphology are consistent with the discovery and speculation of the structure evolution of natisite by washing with water and sulfuric acid solutions (see Section 3.3.1 and Section 3.3.2).

## 4. Conclusions

Natisite (Na_2_TiSiO_5_) with the shape of a square sheet was selectively prepared from TiO_2_ and Na_2_SiO_3_ in NaOH solution by hydrothermal method with 14.3 mol/L NaOH solution at 240 °C, and higher alkali concentration favored its formation. The calculated Raman and IR spectra of natisite from the first-principles calculations agreed well with the experiment, and its vibrational modes were assigned with corresponding atomic displacements. Natisite has 42 active normal modes at ***k*** = 0, including 20 Raman active modes and 22 infrared active modes. The strongest Raman peak at 844 cm^−1^ is assigned to the symmetric stretching of the apical Ti–O bond in the TiO_5_ square pyramids, assigning to the combination of *A*_1g_ and *B*_2g_ modes. Natisite was relatively stable in water, and lower than 6% sodium ions could be leached, while 95.5% Na, 63.4% Si, and 35.2% Ti in total could be leached in further acid washing, indicating the disintegration of natisite structure. During washing in sulfuric acid solutions, the interlayer spacing of natisite was reduced, and an intermediate constructed by SiO_4_ and newly formed TiO_6_ units may be formed. Moreover, treating natisite directly with 40 wt % H_2_SO_4_ solution resulted in the total release of sodium and titanium and silica gel formation.

## Figures and Tables

**Figure 1 materials-14-02259-f001:**
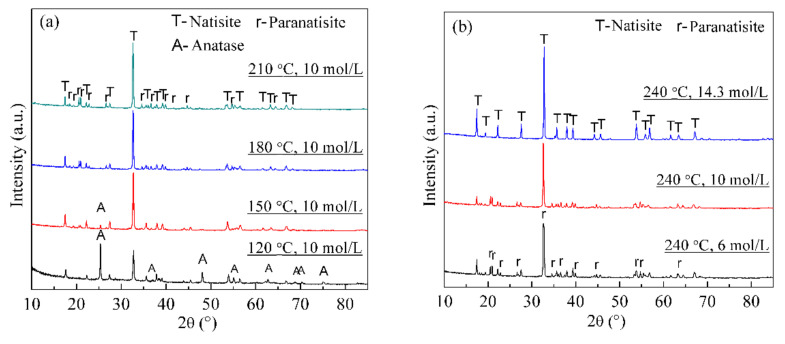
XRD patterns of the hydrothermal products obtained from different temperatures (**a**) and NaOH concentrations (**b**).

**Figure 2 materials-14-02259-f002:**
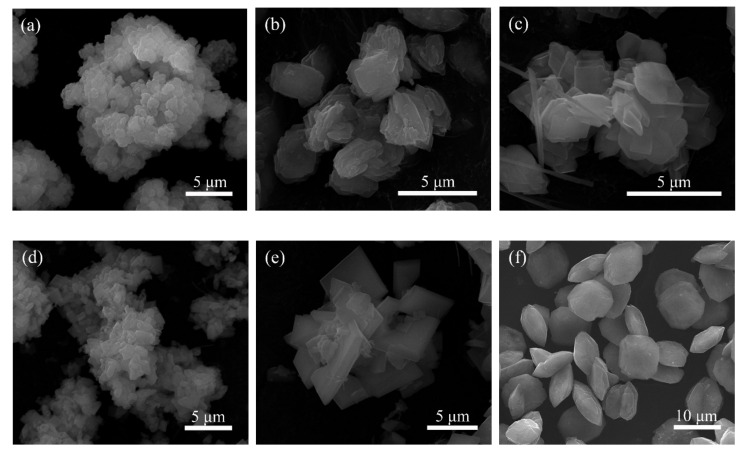
SEM images of the hydrothermal products obtained from different temperatures and NaOH concentrations: (**a**) 150 °C, 10 mol/L; (**b**) 180 °C, 10 mol/L; (**c**) 210 °C, 14.3 mol/L; (**d**) 240 °C, 6 mol/L; (**e**) 240 °C, 10 mol/L; (**f**) 240 °C, 14.3 mol/L.

**Figure 3 materials-14-02259-f003:**
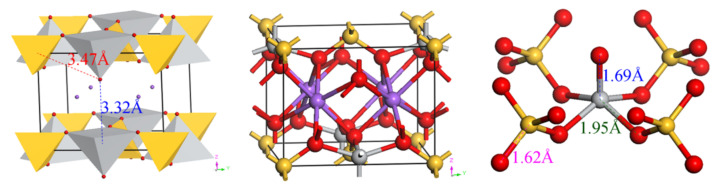
Polyhedral structure and ball-and-stick structure of natisite and O = Ti–(O–Si)_4_ unit. Purple, Na; gray, Ti; yellow, Si; red, O.

**Figure 4 materials-14-02259-f004:**
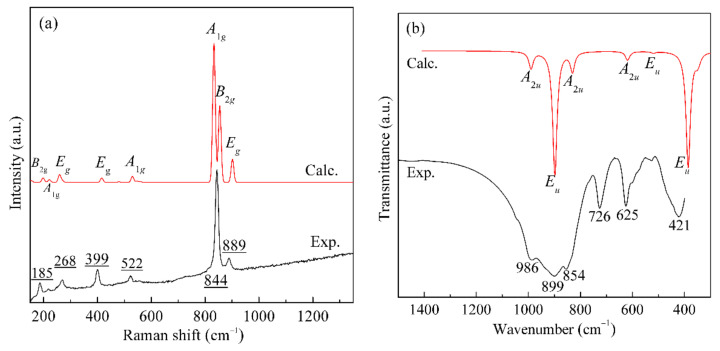
Raman spectra (**a**) and IR spectra (**b**) of natisite from calculation and experiment.

**Figure 5 materials-14-02259-f005:**
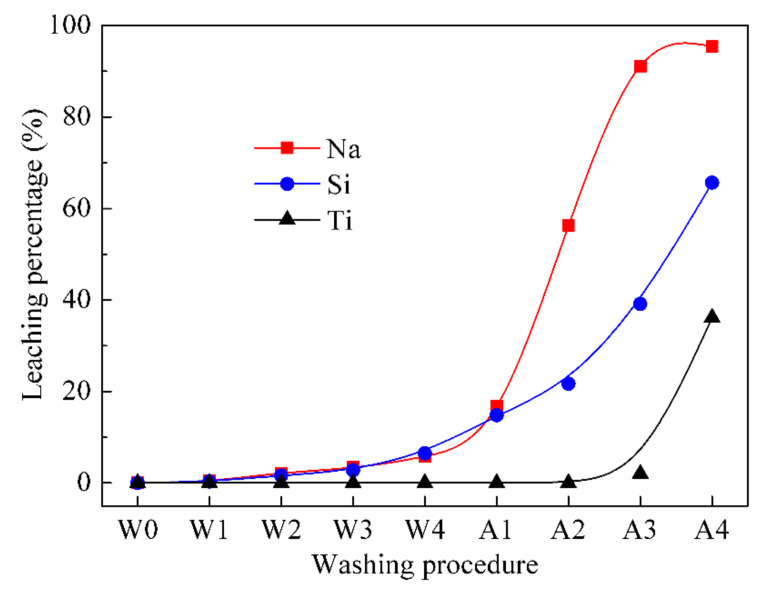
Leaching percentages of Na, Si and Ti from analyzing the solutions of washing natisite with water and sulfuric acid solutions.

**Figure 6 materials-14-02259-f006:**
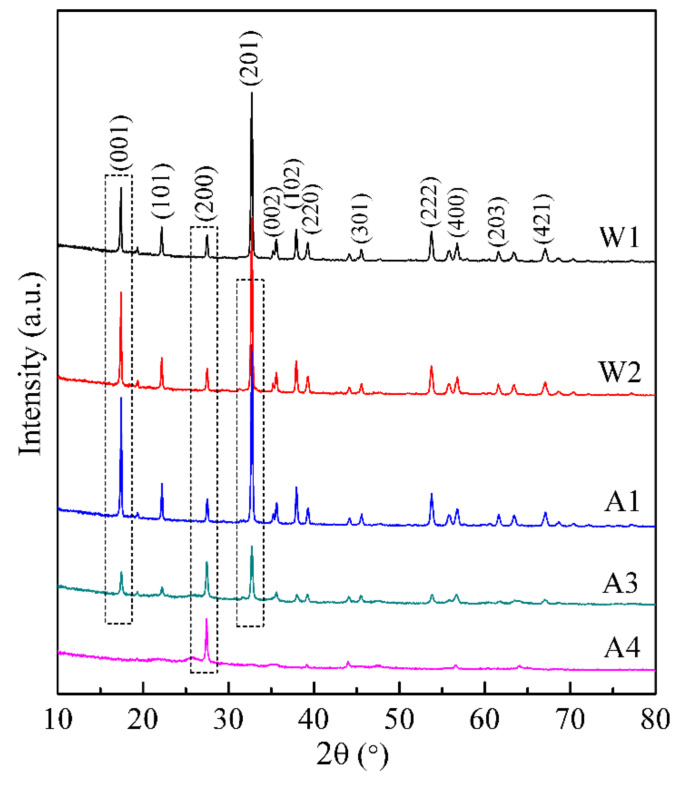
XRD patterns of the solid samples from washing natisite with water and sulfuric acid solutions.

**Figure 7 materials-14-02259-f007:**
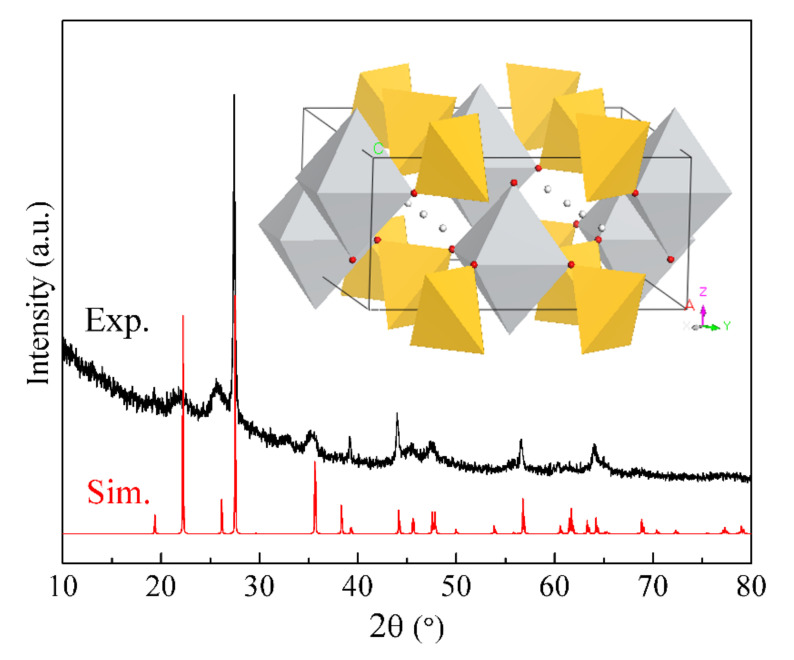
Experimental and simulated XRD patterns of the sample (A4) from washing natisite with sulfuric acid solutions (inset: a possible intermediate structure after Na^+^ in natisite exchanging with H^+^).

**Figure 8 materials-14-02259-f008:**
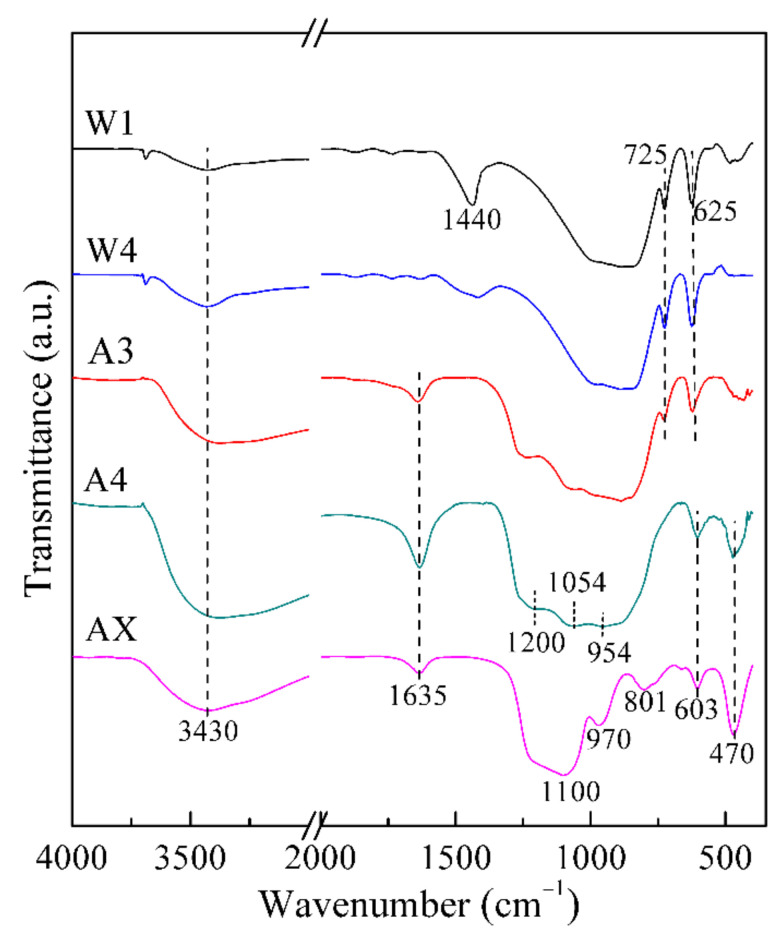
FT-IR spectra of the solid samples from washing natisite with water and sulfuric acid solutions.

**Figure 9 materials-14-02259-f009:**
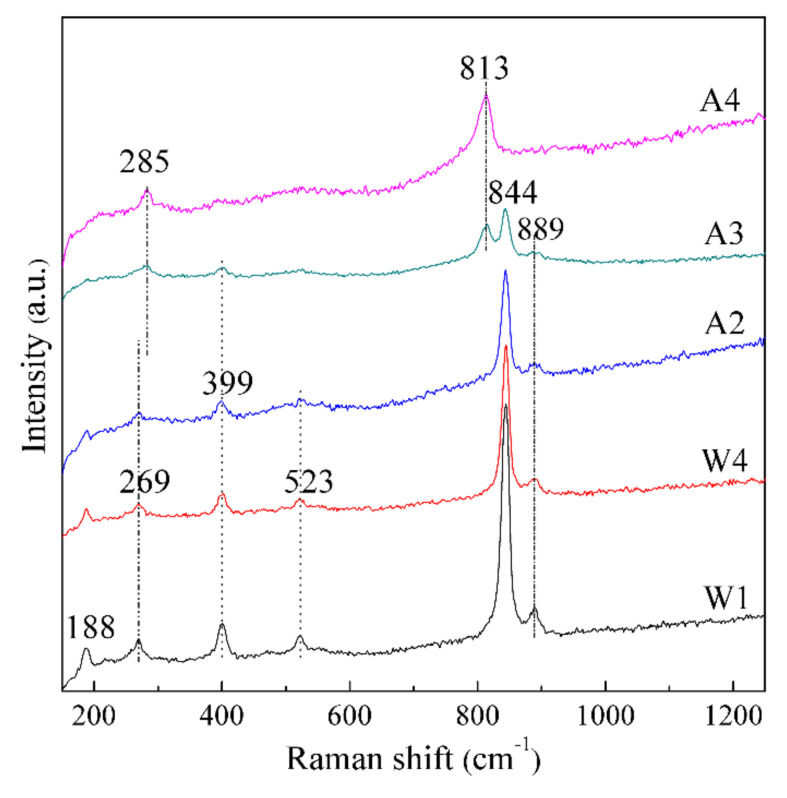
Raman spectra of the solid samples from washing natisite with water and sulfuric acid solutions.

**Figure 10 materials-14-02259-f010:**
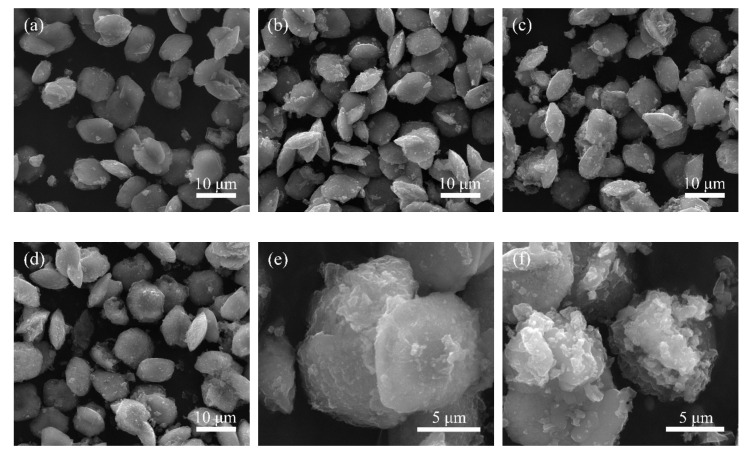
SEM images of the solid samples from washing natisite with water and sulfuric acid solutions. (**a**–**f**) represent the samples W2, W3, W4, A2, A3 and A4, respectively.

**Table 1 materials-14-02259-t001:** Treatment procedures for natisite and names of the obtained solid samples.

Treatment Procedure	Sample
Natisite obtained from a hydrothermal reaction	W0
Wash natisite with water	W1
Wash W1 with water	W2
Wash W2 with water	W3
Wash W3 with water	W4
Wash W4 with H_2_SO_4_ solution, pH = 6.5	A1
Wash A1 with H_2_SO_4_ solution, pH = 4.5	A2
Wash A2 with H_2_SO_4_ solution, pH = 1.0	A3
Wash A3 with H_2_SO_4_ solution, pH = −0.4	A4
Leach W0 with 40 wt % H_2_SO_4_ solution	AX

**Table 2 materials-14-02259-t002:** Atomic site and bond distance of natisite after geometry optimization.

Atom	*x/a*	*y/b*	*z/c*
Ti	0.5000	0.0000	0.9343
O^a^	0.5000	0.0000	0.2714
Na	0.2500	0.2500	0.5000
Si	0.0000	0.0000	0.0000
O^b^	0.0000	0.2087	0.1839
**Atom 1**	**Atom 2**	**Count**	**Distance (Å)**
Ti	O^a^	1x	1.6900
Ti	O^b^	4x	1.9523
Na	O^a^	2x	2.5318
Na	O^b^	4x	2.2650
Si	O^b^	4x	1.6202

Note: O^a^ denotes the apical oxygen in the TiO_5_ unit, and O^b^ denotes the bridging oxygen in the TiO_5_ unit.

**Table 3 materials-14-02259-t003:** Raman and IR frequencies and normal modes of natisite from first-principles calculations.

No.	*v* (cm^−1^)	Irrep.	Active	No.	*v* (cm^−1^)	Irrep.	Active	No.	*v* (cm^−1^)	Irrep.	Active
1	121.2 ^d^	*E_u_*	IR	11	308.5	*A* _2*u*_	IR	20	558.7	*B* _2*g*_	Raman
2	153.8 ^d^	*E_g_*	Raman	12	343.8 ^d^	*E_u_*	IR	21	618.4	*A* _2*u*_	IR
3	178.6	*A* _2*u*_	IR	13	351.4	*A* _2*u*_	IR	22	830.2	*A* _2*u*_	IR
4	195.1 ^d^	*E_u_*	IR	14	385.5 ^d^	*E_u_*	IR	23	833.0	*A* _1*g*_	Raman
5	198.1	*B* _2*g*_	Raman	15	415.7 ^d^	*E_g_*	Raman	24	854.4	*B* _2*g*_	Raman
6	221.2	*A_1g_*	Raman	16	479.7	*B* _1*g*_	Raman	25	897.8 ^d^	*E_u_*	IR
7	248.1 ^d^	*E_u_*	IR	17	518.9 ^d^	*E_u_*	IR	26	900.3	*A* _1*g*_	Raman
8	259.1 ^d^	*E_g_*	Raman	18	530.0	*A* _1*g*_	Raman	27	901.5 ^d^	*E_g_*	Raman
9	269.6 ^d^	*E_g_*	Raman	19	546.4 ^d^	*E_g_*	Raman	28	989.4	*A* _2*u*_	IR
10	288.8 ^d^	*E_u_*	IR								

Note: ^d^ denotes that this frequency has two symmetric vibrations of the same mode and intensity.

## Data Availability

Data sharing is not applicable.

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
