# Peer review of "Vibrational Spectral Analysis of Natisite (Na2TiSiO5) and its Structure Evolution in Water and Sulfuric Acid Solutions"

_materials, 2021, doi:10.3390/ma14092259_

Round 1
Reviewer 1 Report
Novelty: 85%, the presented manuscript is a logical continuation of previous investigations.
Links:
Fancheng Meng, Yahui Liu, Lina Wang, Desheng Chen, Hongxin Zhao, Weijing Wang , Tao Qi, Structural, vibrational, and thermodynamic properties of γ-Na2TiO3: First-principles and experimental studies // Ceramics International Volume 44, Issue 2, 1 February 2018, Pages 2065-2073 https://doi.org/10.1016/j.ceramint.2017.10.152
General recommendations
- Please, add a blank space between temperature numbers and Celsius sign, e.g. ‘772±10 °C’ instead of ‘772±10°C’. Check text, figures and figure captions.
- Check the font size, it differs within one section (especially in Introduction).
- Check font size on figures. E.g. Figure 4: it will be better to reduce the font size of ‘Calc.’ and ‘Exp.’ as well as peak marks.
- Do not begin the new section at the bottom of the page like ‘2. Experiments and Calculations’.
- What is the correct form for mentioning figures in the text? ‘Figure 7’ or ‘Fig. 7’? Please choose one variant and use only it.
- I recommend using either ‘layered structure’ or ‘layered titanosilicate’ instead of ‘layer structure’. But this is up to authors.
Introduction:
‘The second one is paranatisite, which has an orthorhombic structure (Pmc21) with a = 9.181(2), b = 4.800(1), c = 9.811(2), Z = 2 [6]’ – add Å after numbers/
‘from 400 to 650 nm’ – please, it will be better not to separate numbers and measurement units
‘As a layer sodium titanosilicate, it is essential to clarify how much and how Na+ in natisite could be exchanged or removed in water and acid solution, which will also help understand the ion-exchange behavior of natisite as an adsorbent’ – ‘As a layer sodium titanosilicate, it is essential to clarify how much and how Na+ in natisite could be exchanged or removed in water and acid solution, which will also help to understand the ion-exchange behavior of natisite as an adsorbent’
‘CTAB’ – it’s a cetrimonium bromide? Please clarify the abbreviation
‘and the Na2SiO3 was used to as ionic type silicon source’ – ‘and the Na2SiO3 was used as ionic type silicon source’
‘The structure of natisite has been characterized by many spectroscopic methods, such as FT-IR [11, 25], Raman [24, 26], and NMR [1, 24], which gives much detailed structural information’ – ‘The structure of natisite has been characterized by many spectroscopic methods, such as FT-IR [11, 25], Raman [24, 26], and NMR [1, 24], which provide much detailed structural information’
‘Recently, there is growing interest in using the first-principles calculations within density functional theory…’ – ‘Recently, the interest deepens in using the first-principles calculations within density functional theory…’
Experiments and Calculations
‘for 40 min’ – put measurement units in one line with numbers
‘X-ray diffraction (XRD) patterns within the angle range of 5°~90° (2θ)’ – ‘X-ray diffraction (XRD) patterns within the angle range of 5~90° (2θ)’
Results and Discussion
Figure 1: 1) please separate Celsius signs form numbers; 2) do not use on one plot phases and chemical substances, designate A as an anatase and not TiO2
‘molar ratio of Na: Si: Ti’ – unnecessary spaces, change for ‘molar ratio of Na:Si:Ti’
Figure 2: 1) please, make a scale on the images more clear; 2) do not separate the figure on to pages
‘much shorter than 1.90–2.25 Å of Ti–O bonds in anaste and rutile’ – ‘much shorter than 1.90–2.25 Å of Ti–O bonds in anatase and rutile’; please check once again these data: the known Ti–O length for anatase is 1.947 Å, and 1 .959 Å
‘In this study, natisite underwent four times of water washing and acid washing, and the removal of Na, Si, and Ti in the washing process was presented in Figure 7’ – ‘Natisite underwent water and acid washing four times during this study, and the removal of Na, Si, and Ti in the washing process is presented in Figure 7’
‘XRD patterns of the washed residue of natisite are shown in Fig .6.’ – ‘XRD patterns of the washed residue of natisite are shown in Fig. 6.’ – Space should be after dot, and not before
‘Before the disintegration of the layer structure of natisite in aqueous solution, Na+ located between the two layers exchanged with H+ in the solution, and the interlamination distance will be decreased due to the smaller diameter of H+’ – ‘Before the disintegration of the layer structure of natisite in aqueous solution, Na+ located between the two layers exchanges with H+ in the solution, and the interlamination distance decreases due to the smaller diameter of H+’
‘There are two characteristic sharp IR absorption bands of natisite at 725 and 625 cm−1, representing the internal modes of TiO5 and SiO4 structural units, respectively [11], which is consistent with our calculation results (assigning to the A2u and Eu modes)’ – ‘There are two characteristic sharp IR absorption bands of natisite at 725 and 625 cm−1, representing the internal modes of TiO5 and SiO4 structural units, respectively [11], which are consistent with our calculation results (assigning to the A2u and Eu modes)’
‘To further investigate the structure evolution of natisite in a more acidic solution, natitsite was treated with 40 wt.% H2SO4 solution, and the result shows more than 98.3% of Ti was removed, showing a total disintegration of the natisite structure. The infrared spectrum the solid residue shown in Figure 9 indicates that the characteristic absorption peaks of silica gel at 1100, 970, 801 and 470 cm−1 [46].’ – ‘To further investigate the structure evolution of natisite in a more acidic solution, natitsite was treated with 40 wt.% H2SO4 solution, and the result showed that more than 98.3% of Ti had been removed, showing a total disintegration of the natisite structure. The IR spectrum of the solid residue shown in Figure 9 indicates that 1100, 970, 801 and 470 cm−1 absorption peaks are characteristic for silica gel [46].’
‘Ti–O bond shifted to 285 cm−1’ – put measurement units in one line with number
‘During acid washing, the particle surface gradually evolved into floccule, and the regular shape and sharp edges of natisite particles disappear’ – ‘During acid washing, the particle surface gradually evolved into floccule, and the regular shape and sharp edges of natisite particles disappeared’
Figure 11: make a scale more clear.

Author Response
Manuscript ID: materials-1157960
Response to Reviewer #1
General recommendations:
COMMENT (1): Please provide more explanation on the technical and theoretical aspects of this work. For example, provide explanations of what normal modes are, what k=0 means,
Reply: Normal modes are, k=0 means, and Gamma point are all the concept in the subject of solid-state physics. The vibration of normal mode is the simplest and most basic vibration, that is, the simple harmonic vibration of all atoms in a molecule near the equilibrium position with the same frequency and phase position. Any complex vibration in a molecule can be regarded as a linear combination of vibrations of normal mode.
The Bloch function of the electron system has a wave vector k, which is the symbol of the quantum number of the wave function. This wave vector k can be taken in the first Brillouin zone, and its origin (k = 0) is Gamma point (high symmetry point). The calculation of this study is based on the study of real space charge distribution when the system wave function is at the quantum number of Bloch wave vector k = 0 (at Gamma point).
We have cited detailed reference and added the calculation description of Raman and IR spectra in subsection 2.4. (See 1st paragraph in Page 4)
COMMENT (2): Please, add a blank space between temperature numbers and Celsius sign, e.g. ‘772±10 °C’ instead of ‘772±10°C’. Check text, figures and figure captions.
Reply: A blank space is added between temperature numbers and Celsius sign according to your suggestion. It is written as “772±10 °C”. (See Line 6 at 1st paragraph of Introduction in Page 1)
COMMENT (3): Check the font size, it differs within one section (especially in Introduction).
Reply: The font size has been unified, and it is 10 pt in the main text. (see 1st paragraph of Introduction in Page 1 and 4th paragraph of Introduction in Page 2)
COMMENT (4): Check font size on figures. E.g., Figure 4: it will be better to reduce the font size of ‘Calc.’ and ‘Exp.’ as well as peak marks.
Reply: The font size in Figure 4 has been reduced in the revised manuscript. Furthermore, we have checked the font size on all other figures, and the font size in them have been modified. . (see Figure 4 in Page 7)
COMMENT (5): Do not begin the new section at the bottom of the page like ‘2. Experiments and Calculations’.
Reply: The section title of “2. Experiments and Calculations” is not at the bottom of the page alone in the revised manuscript. (see the bottom in Page 2)
COMMENT (6): What is the correct form for mentioning figures in the text? ‘Figure 7’ or ‘Fig. 7’? Please choose one variant and use only it.
Reply: Figure 7 is the correct form for mentioning this figure in the text. We have checked this form throughout the manuscript and modified the unform ones. (see Line 2 at 2nd paragraph in Page 8)
COMMENT (7): I recommend using either ‘layered structure’ or ‘layered titanosilicate’ instead of ‘layer structure’. But this is up to authors.
Reply: Thank you for your careful review. We have used “layered structure” instead of “layer structure” in the revised manuscript. (see Line 1 at 1st paragraph in Page 2)
Introduction:
COMMENT (9): ‘The second one is paranatisite, which has an orthorhombic structure (Pmc21) with a = 9.181(2), b = 4.800(1), c = 9.811(2), Z = 2 [6]’ – add Å after numbers/
‘from 400 to 650 nm’ – please, it will be better not to separate numbers and measurement units
Reply: We have added Å after numbers of a = 9.181(2), b = 4.800(1), and c = 9.811(2). “from 400 to 650 nm” has been modified into “from 400 nm to 650 nm”. (see Line 9 and 14 at 1st paragraph of Introduction in Page 1)
COMMENT (10): ‘As a layer sodium titanosilicate, it is essential to clarify how much and how Na+ in natisite could be exchanged or removed in water and acid solution, which will also help understand the ion-exchange behavior of natisite as an adsorbent’ – ‘As a layer sodium titanosilicate, it is essential to clarify how much and how Na+ in natisite could be exchanged or removed in water and acid solution, which will also help to understand the ion-exchange behavior of natisite as an adsorbent’
Reply: This sentence has been modified as your suggestion. It is written as “As a layered sodium titanosilicate, it is essential to clarify how much and how sodium ions in natisite could be exchanged or removed in water and acid solutions, which is also helpful to understand the ion-exchange behavior of natisite as an adsorbent”. (see Line 14-17 at 2nd paragraph in Page 2)
COMMENT (11): ‘CTAB’ – it’s a cetrimonium bromide? Please clarify the abbreviation
Reply: CTAB is the abbreviation of cetyltrimethyl ammonium bromide, and we have modified it in Page 2. (see Line 6 at 3rd paragraph in Page 2)
COMMENT (12): ‘and the Na2SiO3 was used to as ionic type silicon source’ – ‘and the Na2SiO3 was used as ionic type silicon source’
Reply: “and the Na2SiO3 was used to as ionic type silicon source” has been should be “and the Na2SiO3 was used as ionic type silicon source” according to your suggestion. Besides, this sentence has been deleted due to semantic repetition with another sentence in Introduction.
COMMENT (13): ‘The structure of natisite has been characterized by many spectroscopic methods, such as FT-IR [11, 25], Raman [24, 26], and NMR [1, 24], which gives much detailed structural information’ – ‘The structure of natisite has been characterized by many spectroscopic methods, such as FT-IR [11, 25], Raman [24, 26], and NMR [1, 24], which provide much detailed structural information’
Reply: This sentence has been modified into “The structure of natisite has been characterized by many spectroscopic methods, such as FT-IR [11, 25], Raman [24, 26], and NMR [1, 24], which provide much detailed structural information” as your suggestion. (see Line 1-3 at 4th paragraph in Page 2)
COMMENT (14): ‘Recently, there is growing interest in using the first-principles calculations within density functional theory…’ – ‘Recently, the interest deepens in using the first-principles calculations within density functional theory…’
Reply: This sentence has been written as “Recently, the interest deepens in using the first-principles calculations within density functional theory…” according to your suggestion. (see Line 6-8 at 4th paragraph in Page 2)
Experiments and Calculations
COMMENT (15): ‘for 40 min’ – put measurement units in one line with numbers
‘X-ray diffraction (XRD) patterns within the angle range of 5°~90° (2θ)’ – ‘X-ray diffraction (XRD) patterns within the angle range of 5~90° (2θ)’
Reply: A non-breaking blank space is added between 40 and min to ensure they are in one line. (see Line 2 at 2nd paragraph in Page 3)
We have checked the manuscript throughout and added non-breaking blank space between measurement units and numbers to keep them in one line. ‘X-ray diffraction (XRD) patterns within the angle range of 5~90° (2θ)’ is used in the revised manuscript. (see Line 2-3 at 3rd paragraph in Page 3)
Results and Discussion
COMMENT (16): Figure 1: 1) please separate Celsius signs form numbers; 2) do not use on one plot phases and chemical substances, designate A as an anatase and not TiO2
molar ratio of Na: Si: Ti’ – unnecessary spaces, change for ‘molar ratio of Na:Si:Ti’
Reply: 1) A space has been added between Celsius signs and numbers to separate them in Figure 1. (see Line 5 at 2nd paragraph in Page 4)
2) “A” is designate as an anatase instead of TiO2 in Figure 1. (see Figure 1 in Page 4)
Unnecessary spaces has been deleted, the phrase has been change into “molar ratio of Na:Si:Ti”. (see Line 4 at 1st paragraph in Page 5)
COMMENT (17): Figure 2: 1) please, make a scale on the images more clear; 2) do not separate the figure on to pages
Reply: 1) We have provided a clear scale on the images of Figure 2; 2) To avoid separating the figures on to pages, the second paragraph in subsection 3.1 has been moved after Figure 2. (see Figure 2 in Page 5)
COMMENT (18): ‘much shorter than 1.90–2.25 Å of Ti–O bonds in anaste and rutile’ – ‘much shorter than 1.90–2.25 Å of Ti–O bonds in anatase and rutile’; please check once again these data: the known Ti–O length for anatase is 1.947 Å, and 1 .959 Å
Reply: The known Ti–O lengths are 1.947 Å and 1 .959 Å for anatase, and 1.949 Å and 1.980 Å for rutile. Therefore, this sentence has been modified into “much shorter than 1.94–1.98 Å of Ti–O bonds in anatase and rutile”. (see Line 5 at 1st paragraph in Page 6)
COMMENT (19): ‘In this study, natisite underwent four times of water washing and acid washing, and the removal of Na, Si, and Ti in the washing process was presented in Figure 7’ – ‘Natisite underwent water and acid washing four times during this study, and the removal of Na, Si, and Ti in the washing process is presented in Figure 7’
Reply: In combination with your and another reviewer’ suggestions, the above sentence has been written as “In this study, natisite was washed with water for four times, then washed with sulfuric acid solution for four times as described in subsection 2.2. The leaching percentages of Na, Si, and Ti in the washing process are presented in Figure 5”. (see Line 1-3 at 2nd paragraph in Page 7)
COMMENT (20): ‘XRD patterns of the washed residue of natisite are shown in Fig .6.’ – ‘XRD patterns of the washed residue of natisite are shown in Fig. 6.’ – Space should be after dot, and not before
Reply: Figure 6 is the correct form for mentioning this figure in the text. (see Line 2 at 2nd paragraph in Page 8)
COMMENT (21): ‘Before the disintegration of the layer structure of natisite in aqueous solution, Na+ located between the two layers exchanged with H+ in the solution, and the interlamination distance will be decreased due to the smaller diameter of H+’ – ‘Before the disintegration of the layer structure of natisite in aqueous solution, Na+ located between the two layers exchanges with H+ in the solution, and the interlamination distance decreases due to the smaller diameter of H+’
Reply: This sentence has been modified into “Before the disintegration of the layered structure of natisite in sulfuric acid solution, Na+ located between the two layers exchanges with H+ in the solution, and the interlamination distance decreases due to the smaller diameter of H+”. (see Line 1-3 at 2nd paragraph in Page 9)
COMMENT (22): ‘There are two characteristic sharp IR absorption bands of natisite at 725 and 625 cm−1, representing the internal modes of TiO5 and SiO4 structural units, respectively [11], which is consistent with our calculation results (assigning to the A2u and Eu modes)’ – ‘There are two characteristic sharp IR absorption bands of natisite at 725 and 625 cm−1, representing the internal modes of TiO5 and SiO4 structural units, respectively [11], which are consistent with our calculation results (assigning to the A2u and Eu modes)’
Reply: This sentence has been moved to subsection 3.1, and it is written as “There are two characteristic sharp IR absorption bands at 726 and 625 cm−1, representing the internal modes of TiO5 and SiO4 structural units, respectively [11], which are consistent with our calculation result (assigning to the A2u and Eu modes)”. (see Line 5-8 at 1st paragraph in Page 7)
COMMENT (23): ‘To further investigate the structure evolution of natisite in a more acidic solution, natitsite was treated with 40 wt.% H2SO4 solution, and the result shows more than 98.3% of Ti was removed, showing a total disintegration of the natisite structure. The infrared spectrum the solid residue shown in Figure 9 indicates that the characteristic absorption peaks of silica gel at 1100, 970, 801 and 470 cm−1 [46].’ – ‘To further investigate the structure evolution of natisite in a more acidic solution, natitsite was treated with 40 wt.% H2SO4 solution, and the result showed that more than 98.3% of Ti had been removed, showing a total disintegration of the natisite structure. The IR spectrum of the solid residue shown in Figure 9 indicates that 1100, 970, 801 and 470 cm−1 absorption peaks are characteristic for silica gel [46].’
Reply: In combination with another reviewer’ suggestion, this sentence has been changed, and Figure 9 has been included in Figure 8. This sentence is written as “In another treatment process, natisite was directly leached with 40 wt.% H2SO4 solution, and almost all sodium and titanium in natisite could be leached. The FT-IR spectrum of its solid residue (sample AX) in Figure 8 exhibits the characteristic absorption peaks of silica gel at 1100, 970, 801 and 470 cm−1 [46], indicating a total disintegration of natisite structure and the formation of silica gel in this process”. (see the last paragraph in Page 9 and first paragraph in Page 10)
COMMENT (24): ‘Ti–O bond shifted to 285 cm−1’ – put measurement units in one line with number
Reply: A non-breaking blank space is added between 285 and cm−1 to ensure they are in one line. (see Line 6-7 at 2nd paragraph in Page 10)
COMMENT (25): ‘During acid washing, the particle surface gradually evolved into floccule, and the regular shape and sharp edges of natisite particles disappear’ – ‘During acid washing, the particle surface gradually evolved into floccule, and the regular shape and sharp edges of natisite particles disappeared’
Reply: This sentence has been modified into “During acid washing, the particle surface gradually evolved into floccule, and the regular shape and sharp edges of natisite particles disappeared”. (see Line 3-4 at 2nd paragraph in Page 11)
COMMENT (26): Figure 11: make a scale more clear.
Reply: We have provided a clearer scale on the images of Figure 10 (Figure 11 in the previous manuscript). (see Figure 2 in Page 5)
Thank you very much for your careful review and constructive suggestions.

Reviewer 2 Report
The article discussed the synthesis and analysis of Natistite with an emphasis on Raman and infrared spectroscopy. The article presents a comprehensive study including computational and experimental results.
The article, however, assumes that the reader is highly experienced with vibrational spectroscopy for material discovery. I would therefore suggest to revise the article to make it easier to understand for a broader readership. After that, I believe the article will be a very valuable contribution to Materials.
Mayor comments:
- Please provide more explanation on the technical and theoretical aspects of this work. For example, provide explanations of what normal modes are, what k=0 means, what the Gamma point is, etc. “However, the vibrational modes of Raman peaks still remain unassigned, and there are several contradictory ideas between different literatures [24, 26].” Please specify what is contraditory. Provide detailed descriptions on how the Raman spectra and IR spectra were calculated, reference software and equations where possible.
- Page 2: “...among which natisite is a more stable phase in hydrothermal conditions”. There is no unique understanding what hydrothermal conditions are. Please specify.
- Figure 6: Please think of a way to present this figure in a way that it allows to compare the XRD patterns. Maybe with a zoomed insert with patterns superimposed? Also mention the assumptions to calculate the XRD pattern, and mention the affect particle morphology can have as this was not mentioned when comparing the experimental XRD patterns.
Minor comments:
- Parts of the introduction are repetitive. E.g. “In this study, natisite with the shape of square sheet was selectively prepared from hydrothermal method with Na2SiO3 as the silicon source.” Is mentioned (in this or another way) several times.
- The font size seems to change several times in the manuscript.
- For the TiO2 precursor (with XRD patterns provided) the phase (anatase, rutile, etc.?) should be specified and referenced to the literature.
- Please provide more details when discussing the synthesis. The stirring speed is provided when describing the synthesis. This is of no use, however, if details on the solution volume, and beaker and stir bar geometry are missing. In general the particle size of precursors should be mentioned. Was an inert atmosphere required to synthesise the material? What was the heating rate used to synthesise the material? What was the pH of the solution used for the washing steps?
Author Response
Manuscript ID: materials-1157960
Response to Reviewer #2
Major comments:
COMMENT (1): Please provide more explanation on the technical and theoretical aspects of this work. For example, provide explanations of what normal modes are, what k=0 means, what the Gamma point is, etc. “However, the vibrational modes of Raman peaks still remain unassigned, and there are several contradictory ideas between different literatures [24, 26].” Please specify what is contraditory. Provide detailed descriptions on how the Raman spectra and IR spectra were calculated, reference software and equations where possible.
Reply: Normal modes are, k=0 means, and Gamma point are all the concept in the subject of solid-state physics. The vibration of normal mode is the simplest and most basic vibration, that is, the simple harmonic vibration of all atoms in a molecule near the equilibrium position with the same frequency and phase position. Any complex vibration in a molecule can be regarded as a linear combination of vibrations of normal mode.
The Bloch function of the electron system has a wave vector k, which is the symbol of the quantum number of the wave function. This wave vector k can be taken in the first Brillouin zone, and its origin (k = 0) is Gamma point (high symmetry point). The calculation of this study is based on the study of real space charge distribution when the system wave function is at the quantum number of Bloch wave vector k = 0 (at Gamma point).
The infrared absorption intensities are related to the dynamical (Hessian) matrix and to the Born effective charges, also known as atomic polarizability tensors, and can be obtained by calculating the phonons at the Gamma point. The Raman spectrum is based on the Raman shift of inelastic scattering of monochromatic light. We have added a description and a reference on how to calculate the Raman spectra and IR spectra in the manuscript in the revised manuscript. (see Line 13-15 at 1st paragraph in Page 4)
We have rechecked the two literature, and found that there is no contradictory, because they discussed the Raman peaks of natisite at different regions. Reference [24] assigned the most prominent bands near 850 cm−1 for natisite to the Ti–Oa bond (Oa denotes an apical oxygen). Reference [26] speculated that the peak related to the Si–O stretching in SiO4–TiOn linkages was positioned at near 900 cm−1 for natisite. However, they only discussed the vibration of Ti–O or Si–O bonds of the Raman peaks at around 850 cm−1 and 900 cm−1, which are not complete and need further verification. In this case, this sentence has changed into “Previous studies only speculated the vibration of Ti‒O and Si‒O bonds of the Raman peaks at around 850 cm−1and 900 cm−1, which are not are not complete and need further experimental or theorical verification. Therefore, the vibrational modes of Raman peaks of natisite still remain unassigned”. (see Line 3-5 at 4th paragraph in Page 2)
COMMENT (2): Page 2: “...among which natisite is a more stable phase in hydrothermal conditions”. There is no unique understanding what hydrothermal conditions are. Please specify.
Reply: The hydrothermal conditions is specified in the manuscript. The above sentence has been modified into “...among which natisite is a more stable phase in NaOH solutions at ≥ 200°C”. (see Line 20 at 1st paragraph of Introduction in Page 1)
COMMENT (3): Figure 6: Please think of a way to present this figure in a way that it allows to compare the XRD patterns. Maybe with a zoomed insert with patterns superimposed? Also mention the assumptions to calculate the XRD pattern, and mention the affect particle morphology can have as this was not mentioned when comparing the experimental XRD patterns.
Reply: Figure 6(a) is XRD patterns of the samples from washing natisite with water and sulfuric acid solutions. To better compare the XRD patterns, we have drawn three rectangular frames with short dash of different colours in Figure 6, which mark the main change of the XRD peaks at crystal planes (001), (200) and (201).
Figure 6(b) is the simulated XRD patterns of natisite obtained by replacing Na+ in its crystal structure with H+ in the different portions, which is calculated in Materials Studio software 6.0. However, the previous calculation assumes that the structure of natisite does not change with the removal of Na+ during washing with water and sulfuric acid solutions, which is not consistent with the fact. Therefore, Figure 6b is deleted from the revised manuscript.
(see Figure 6 in Page 8)
Minor comments:
COMMENT (4): Parts of the introduction are repetitive. E.g. “In this study, natisite with the shape of square sheet was selectively prepared from hydrothermal method with Na2SiO3 as the silicon source.” Is mentioned (in this or another way) several times.
Reply: Such repetitive sentence in the third paragraph of the introduction has been deleted. To summarize the references of preparing natisite, another sentence has been added at the end of this paragraph, which is “Previous studies have shown that hydrothermal method is a reasonable approach to prepare natisite with controllable morphology”. (see Line 6-8 at 3rd paragraph in Page 2)
COMMENT (5): The font size seems to change several times in the manuscript.
Reply: The font size has been unified, and it is 10 pt in the main text. (see 1st paragraph of Introduction in Page 1 and 4th paragraph of Introduction in Page 2)
COMMENT (6): For the TiO2 precursor (with XRD patterns provided) the phase (anatase, rutile, etc.?) should be specified and referenced to the literature.
Reply: TiO2 precursor used in this study is anatase phase with the particle size of 0.2~0.3 μm. This part has been re-written as below: “NaOH (purity ≥98.0%,), TiO2 (purity ≥99.0%, anatase phase, particle size 0.2~0.3 μm) and Na2SiO3·9H2O (purity ≥98.0%) and deionized water were used to prepare natisite by hydrothermal method. All these reagents were analytically pure and purchased from Sinopharm Chemical Reagent Co., Ltd”. (see Line 1-6 at 6th paragraph in Page 2)
COMMENT (7): Please provide more details when discussing the synthesis. The stirring speed is provided when describing the synthesis. This is of no use, however, if details on the solution volume, and beaker and stir bar geometry are missing. In general the particle size of precursors should be mentioned. Was an inert atmosphere required to synthesise the material? What was the heating rate used to synthesise the material? What was the pH of the solution used for the washing steps?
Reply: More details have been provided when discussing the synthesis in the revision. It is true that providing the stirring speed without mentioning the solution volume, and beaker and stir bar geometry is no use. This sentence has been changed into “Thereafter, the mixture was poured into a nickel autoclave and sealed, which is mechanically stirred to keep it in suspension”. (see Line 1-2 at 1st paragraph in Page 3)
Natisite is synthesized at autogenous pressure of the system, and no inert atmosphere is required. The particle size of TiO2 precursor is 0.2~0.3 μm. (see Line 1 at 6th paragraph in Page 2)
The autoclave was heated to the selected temperature at a heating rate of 5°C/min. (see Line 3-4 at 1st paragraph in Page 3)
The pH of the solution in washing natisite with water was not detected, it should be above 7.0. After water washing, it was washed with the dilute sulfuric solutions four times with the pH values being kept to 6.5, 4.5, 1.0, and -0.4, respectively. (see Line 1-5 at 2nd paragraph in Page 3)
Thank you very much for your careful review and constructive suggestions.

Reviewer 3 Report
See the attached PDf.

Author Response
Manuscript ID: materials-1157960
Response to Reviewer #3
COMMENT (1): This research is quite interesting and of interest to the readers, but some parts of the manuscript reveal lack of proficiency in English language. Therefore, some phrases need revision by an expert in English language.
For example, the current title should be ameliorated. Instead of "... Structure Evolution by Treatment of Water and Acid Solution"', it could be:"... Structure Evolution in Aqueous and Sulfuric Acid Solutions". Also, along the text, the expression "by the treatment of water and acid solution" should be replaced by a more suitable technical English expression. When writing "the treatment of acid solution" (see the Abstract), the authors wish to mean "treatment with acid solution".
Reply: Thank you very much for your careful review. We have improved the English language of this manuscript by an expert in English language. Your revision suggestion of the title is reasonable. Considering the actual treatment of natisite, water was used to wash natisite to investigate its behaviour in water, thus "... Structure Evolution in Water and Sulfuric Acid Solutions" is a clear expression. (see the Title in Page 1)
For the same reason, "by the treatment of water and acid solution" has been changed in to “washing with sulfuric acid solutions” in the Abstract. (see Line 9 in Abstract in Page 1)
COMMENT (2): Reconsider the Keywords. Do not use "preparation" (Preparation of what?). Instead of "Raman", it should be "Raman spectroscopy"'. Include "IR spectroscopy".
Reply: The Keywords have been reconsidered and changed. The Keywords in the revision are “natisite; vibration spectroscopy; structure evolution; sodium ion; aqueous solution”. (see Keywords in Page 1)
COMMENT (3): Please note that in the Abstract and in the Conclusions, you have mentioned that "Natisite has... 22 infrared active modes...". However, (contrarily tothe 20 Raman active modes) the 22 infrared active modes are not mentioned anywhere else. Therefore, the readers will question the reason for this statement about the "22 infrared active modes" of natisite.
Reply: The original manuscript focused on Raman spectrum, and the infrared analysis was omitted. In the revised manuscript, the statement of infrared spectrum and a new figure (Figure 5) showing that the calculated and experimental infrared spectra of natisite have been added.
It is written as below in subsection 3.2:
“Table 3 presents the predicted vibration frequencies of normal modes with the respective irreducible representations and the assignment of IR and Raman active modes of natisite obtained from the first-principles calculations. The point group of natisite is D4h, and there are 20 Raman active modes and 22 IR active modes according to the group theory analysis of eigenvectors with GRaman = 12Eg + 4A1g + 3B2g + B1g and GIR = 16Eu + 6A2u. However, there are only 14 Raman frequencies and 14 IR frequencies due to the existence of six Eg modes and eight Eu modes corresponding to two symmetric vibrations at the same frequency”. (see Line 1-8 at 2nd paragraph in Page 6)
“Figure 6 shows the shows that the calculated infrared spectrum also matches well with the experimental one of natisite. The absorption bands at 986, 899, and 854 cm-1 are attributed to the symmetric stretching vibration of Si-O bond (A2u mode), the antisymmetric stretching vibration of Si-O bond (Eu mode), and the stretching of the apical Ti–O bond in the TiO5 square pyramids, respectively. There are two characteristic sharp IR absorption bands at 726 and 625 cm-1, representing the internal modes of TiO5 and SiO4 structural units, respectively, which is consistent with our calculation result (assigning to the A2u and Eu modes). The antisymmetric stretching vibration of Ti-Ob bonds (Eu modes) may lead to the IR absorption at 421 cm-1”. (see Line 1-9 at 1st paragraph in Page 7)
COMMENT (4): In page 2, the abbreviation CTAB should be disclosed to the readers.
Reply: CTAB is the abbreviation of cetyltrimethyl ammonium bromide, and we have modified it in Page 2. (see Line 6 at 3rd paragraph in Page 2)
COMMENT (5): In page 2 (at the end of the last paragraph, just before section 2. Experiments and Calculations), instead of "systemically"', write "systematically".
PLEASE NOTE that Section 2. Experiments and Calculations is an important part of this manuscript. In a scientific paper, all procedures followed by the authors should be clearly described. It is very important that readers are informed about all experimental details so that they can repeat the experiments if they are in possession of identical materials and equipment. In its present form, even though some procedures are well explained, the manuscript does not correctly describe all experimental details.
Reply: "systemically" has been changed into "systematically". (see the end of 5th paragraph in Page 2)
We have added more experimental details in this part to make all procedures clear and easy to repeat, and a new table that list the serial treatment procedures for natisite and the names of the obtained samples are provided. The subsections 2.1 and 2.2 have been rewritten in the revised manuscript. (see subsections 2.1and 2.2 in Page 1 and 2)
COMMENT (6): At the end of subsection 2.1. (Sample preparation) provide the type of alcohol used ... and, also, describe the procedure used for drying the solid product.
Reply: We have added more experimental details in the post-treatment of the solid product. The solid product was sampled, washed with analytically-pure ethanol, and dried at 80°C in an electric blast drying oven overnight for further analysis. (see Line 6-8 at 1st paragraph in Page 3)
COMMENT (7): As it is, the long paragraph in subsection 2.2. is very puzzling for the readers to follow. The expression "in a compared test" is confusing. For the sake of clarity and even for the choice of a good title for the manuscript, it is essential that the treatments carried out in this research work are dully presented/described so that other researchers can repeat them. Forthat, I suggest that a table is inserted in subsection 2.2. (Na+ removing tests...). In this new table, the various steps/treatments (WO, Wl, W2, W3, W4, Al, A2, A3, A4, "treatment with 40 wt.% sulfuric solution") should be adequately described (including all details: temperature, duration, pH values, etc) so that no doubts become raised/left to the readers.
Note that only in Figure 5 and in Figure 11 the samples were named WO, Wl, W2, W3, W4, Al, A2, A3 and A4. The readers do not know the exact/full meaning of each of these sample codes.
Reply: The serial treatment procedures for natisite and the names of the obtained samples are listed in Table 1. (see 2nd paragraph and Table 1 in Page 3)
Table 1. Treatment procedures for natisite and sample names.
|
Treatment procedure |
Sample |
|
natisite prepared from hydrothermal reaction |
W0 |
|
wash natisite with water |
W1 |
|
wash W1 with water |
W2 |
|
wash W2 with water |
W3 |
|
wash W3 with water |
W4 |
|
wash W4 with H2SO4 solution, pH = 6.5 |
A1 |
|
wash A1 with H2SO4 solution, pH = 4.5 |
A2 |
|
wash A2 with H2SO4 solution, pH = 1.0 |
A3 |
|
wash A3 with H2SO4 solution, pH = −0.4 |
A4 |
|
leach natisite with 40 wt.% H2SO4 solution |
AX |
COMMENT (8): At the beginning of subsection 2.4., disclose the meaning of the abbreviation DFT. This can be done, for example, by writing: "The CASTEP package was used to carry out the plane-wave pseudopotential calculations using density functional theory (DFT) [30]".
Reply: Thank you for your suggestion. This sentence has been changed into "The CASTEP package was used to carry out the plane-wave pseudopotential calculations using density functional theory [30]". (see Line 1-2 at 1st paragraph in Page 4)
COMMENT (9): In subsection 2.4., explain to the readers the exact meaning of the phrase: "The default setting of ultrafine was used to perform the geometry optimization".
Reply: The default setting of ultrafine used to perform the geometry optimization was the plane wave basis set cut-off energy was 830 eV with the total energy convergence tolerance of 5×10−6 eV/atom. This setting was provided later in the same paragraph. Therefore, this sentence is deleted in the revised manuscript. (see Line 12-13 at 1st paragraph in Page 4)
COMMENT (10): At the beginning (4th line) of subsection 3.1. (Selective Preparation of Natisite) it is mentioned that "TiO2 disappears gradually with the increase of temperature in 6M NaOH solution", but then Figure la shows only the results obtained with 10M NaOH solution.
Reply: “M” is replaced by “mol/L”, since “M” is not the UI unit. It should be “TiO2 disappears gradually with the increase of temperature in 10 mol/L NaOH solution”. (see Line 5 at 2nd paragraph in Page 4)
COMMENT (11): At the 2nd paragraph of subsection 3.1. (Selective Preparation of Natisite) it is mentioned that "As shown in Figure 2a-2c, the morphologies of the products become more regular in 6M NaOH solution with the increase of reaction temperature, few fiber materials were formed at 210°C, and the lamellar aggregates were obtained at 240°C ", but then only one SEM image of natisite obtained with 6M NaOH concentration is shown in Figure 2.
Reply: We are sorry for the mistake. It should be written as “As shown in Figure 2, the morphologies of hydrothermal products become more regular with the increase of reaction temperature in 10 mol/L NaOH solution, few fibrous materials were formed at 210 °C, and the lamellar aggregates were obtained at 240 °C”. There are four SEM images of hydrothermal products from 10 mol/L NaOH solution in Figure 2, and they are Figure 2a, Figure 2b, Figure 2c, and Figure 2e. (see Line 1-4 at 3rd paragraph in Page 4)
COMMENT (12): Instead of "anaste", write "anatase" (mid of page 5).
Reply: This is a misspelling. It should be written as “anatase”, and we have modified it in the revised manuscript. (see Line 6 at 1st paragraph in Page 6)
COMMENT (13): (Page 6,1st paragraph): Figure 6 is mentioned here, but it should be Figure 4 (Raman spectra of natisite from calculation and experiment).
Reply: It should be Figure 4 here, not Figure 6. We have modified it in the revised manuscript. (see Line 1 at 3rd paragraph in Page 6)
COMMENT (14): (Page 6,2nd paragraph): Instead of "The Raman peak at 899 cm-1...", it should be "The Raman peak at 889 cm"1...".
Reply: We are very sorry for the mistake. It should be 889 cm−1, instead of 899 cm−1. We have modified it in the revised manuscript. (see Line 2 at 3rd paragraph in Page 6)
COMMENT (15): Table 3 (shown in page 7) should be correctly numbered (taking into consideration the other previous tables). Additionally, since the data is divided in 3 groups, perhaps it is better to draw vertical lines to separate them.
Reply: Table 3 (shown in page 7) should be numbered as Table 2 in the previous manuscript, and it was our mistake. Since a new Table has been added before to provide the treatment details of natisite, it is still Table 3 in the revised manuscript. Vertical lines have been drawn to the three groups of data. (see Table 3 in Page 7)
COMMENT (16): (1st line at 1st paragraph of subsection 3.3.1.) Reconsider the phrase "natisite underwent four times of water washing and acid washing". And reconsider also the next sentence "and the removal of Na, Si, and Ti in the washing process was presented in Figure 7". It should be "is" (instead of "was") and "Figure 5" (instead of "Figure 7").
Reply: We are very sorry for the mistakes. The first two lines at 1st paragraph of subsection 3.3.1 has been written as “In this study, natisite was washed with water for four times, then washed with sulfuric acid solution for four times as described in subsection 2.2. The leaching percentages of Na, Si, and Ti in the washing process are presented in Figure 5”. (see Line 1-3 at 2nd paragraph in Page 7)
COMMENT (17): Caption of Figure 5 should be ameliorated mentioning the technique used to obtain the results plotted in the figure.
Reply: The results plotted in Figure 5 was obtained by analyzing the solution from washing natisite with water and sulfuric acid solution, and ICP-OES was used to analyze the composition of the solution. Therefore, the caption of Figure 5 has been modified into “Figure 5. Leaching percentages of Na, Si, and Ti from analyzing the solution of washing natisite with water and sulfuric acid solutions”. (See Figure 5 in Page 8)
COMMENT (18): Caption of Figure 6 should be amended mentioning the difference between (a) and (b) parts, and providing information on the meaning of 0%, 10%, 30%, 50% and 70% in (b).
Reply: Figure 6(a) is XRD patterns of the samples from washing natisite with water and sulfuric acid solutions. To better compare the XRD patterns, we have drawn two rectangular frames with short dash of different colours in Figure 6, which mark the main change of the XRD peaks at 17.36° and 27.46°. Figure 6(b) is the simulated XRD patterns of natisite obtained by replacing Na+ in its crystal structure with H+ in the different portions, which is calculated in Materials Studio software 6.0. However, this calculation is based on the assumption that the structure of natisite does not change with the removal of Na+ during washing with water and sulfuric acid solutions, which is not consistent with the fact. Therefore, Figure 6b is deleted from the revised manuscript, and the caption of Figure 6 is “XRD patterns of the samples from washing natisite with water and sulfuric acid solutions.” (See Figure 6 in Page 8)
COMMENT (19): At the end of the last paragraph of page 8, figure numbers of XRD pattern and Raman spectrum must be indicated.
Reply: According to Figure S1 in the Supplementary Materials, its XRD pattern is indicate as Figure S1(a), and Raman spectrum is indicated as Figure S1b. Therefore, this sentence has been changed into “its XRD pattern Figure S1a and Raman spectrum in Figure S1b both show that the calcination product belongs to anatase.” (see Line 4 at 1st paragraph in Page 9)
COMMENT (20): At the last paragraph in page 10, reconsider/reformulate the phrase: "To further investigate the structure evolution of natisite in a more acidic solution, natisite was treated with 40 wt.% H2SO4 solution, and the result shows more than 98.3% of Ti was removed, showing...". Suggestion: "As mentioned in subsection 2.2., to further investigate the structure evolution of natisite in a more acidic solution, natisite was treated with 40 wt.% H2SO4 solution (H2SO4/Na2TiSiOs molar ratio of 3:1, at 65°C for 4 hours). The result shows that more than 98.3% of Ti was removed, leading to total disintegration of the natisite structure".
Reply: According to your comment #5 and #7, the details of treating natisite directly in a more acidic solution are added in subsection 2.2 and Table 1 in the revised manuscript, the previous Figure 9 has been included in Figure 8. (see Figure 8 in Page 10)
This sentence has been written as “In another treatment process, natisite was directly leached with 40 wt.% H2SO4 solution, and almost all sodium and titanium in natisite could be leached. The FT-IR spectrum of its solid residue (sample AX) in Figure 8 exhibits the characteristic absorption peaks of silica gel at 1100, 970, 801 and 470 cm−1 [46], indicating a total disintegration of natisite structure and the formation of silica gel in this process”. (see the last paragraph in Page 9 and the first paragraph in Page 10)
COMMENT (21): Caption of Figure 9 should be amended mentioning the details of the treatment. Suggestion: "Figure 9. FT-IR spectrum of the residue from direct leaching of natisite with the 40 wt.% H2SO4 solution (H2SO4/Na2TiSiOs molar ratio of 3:1, at 65°C for 4 hours)".
Reply: The details of the treatment is provided in Table 1 in the revised manuscript. Moreover, this figure is deleted, and the FT-IR spectrum of the residue from direct leaching of natisite is included in Figure 8 in the revised manuscript. (see Figure 8 in Page 10)
COMMENT (22): (Last line at the paragraph in page 11): Please explain what you mean by "... the intermediate in Figure 7".
Reply: The original statement is confusing, and this sentence has been changed into “the Raman vibration peak at 813 cm−1 just falls within the vibration range of Ti–O bond in the end point of TiO6 octahedron, which may prove the existence of TiO6 unit as shown in the intermediate structure in Figure 7.” (see Figure 5-7 in Page 11)
COMMENT (23): (1st line at the 1st paragraph in page 12): Reconsider/reformulate the phrase: "The morphological changes of natisite of removing Na* during water and acid washing were compared and studied in Figure 11". Suggestion: "The morphological changes of natisite due to Na* removal during water and acid washing are compared in Figure 11".
Reply: This sentence is now written as " The morphological changes of natisite in water and sulfuric acid solutions were compared, as shown in Figure 10". (see Line 1-2 at 2nd paragraph in Page 11)
COMMENT (24): (Last phrase at the 1st paragraph in page 12, before Conclusions): Reconsider/reformulate the phrase. Suggestion: "The above macroscopic changes in morphology are consistent with the discovery and speculation of the structure evolution of natisite during Na+ ions removal (see subsections 3.3.1 and 3.3.2.)".
Reply: This sentence is now written as " The above macroscopic changes in morphology are consistent with the discovery and speculation of the structure evolution of natisite by washing with water and sulfuric acid solutions (see subsections 3.3.1 and 3.3.2)". (see Line 7-9 at 2nd paragraph in Page 11)
Thank you very much for your careful review and constructive suggestions.

Round 2
Reviewer 3 Report
The authors satisfactorily addressed my comments.
Nonetheless, the phrase “Furthermore, the Raman vibration peak at 813 cm−1 just falls within the vibration range of Ti–O bond in the end point of TiO6 octahedron, which may prove the formation of TiO6 unit as shown in the intermediate structure in Figure 7” must be amended. Instead of Figure 7, I think it should be Figure 9. And, additionally please note that the expression “in the intermediate structure of” is confusing for most of the readers. What is an “intermediate structure” in a graph showing Raman spectra?
Author Response
Response to Reviewer #3
COMMENT (1): Nonetheless, the phrase “Furthermore, the Raman vibration peak at 813 cm−1 just falls within the vibration range of Ti–O bond in the end point of TiO6 octahedron, which may prove the formation of TiO6 unit as shown in the intermediate structure in Figure 7” must be amended. Instead of Figure 7, I think it should be Figure 9. And, additionally please note that the expression “in the intermediate structure of” is confusing for most of the readers. What is an “intermediate structure” in a graph showing Raman spectra?
Reply: Thank you very much for your careful review. I am sorry for the confusing expression. The Raman peaks attributed to the vibration in TiO6 octahedron in Figure 9 is consistent with the speculation and discussion of the formation of an intermediate structure containing TiO6 unit in subsection 3.3.1, which is shown in Figure 7. To make it clearer, this sentence has been changed into “which is consistent with the speculation of the formation of an intermediate structure containing TiO6 units in subsection 3.3.1”. (see Line 6-7 at 1st paragraph in Page 11)
Furthermore, the English of this manuscript is checked and some expressions have been improved. Abbreviations of some journals are also substituted in the reference list.
Thank you very much for your careful review and constructive suggestions.
